# Development and Validation of a New Tool in Predicting In-Hospital Mortality for Hip-Fractured Patients: The PRIMOF Score

**DOI:** 10.3390/medicina58081082

**Published:** 2022-08-11

**Authors:** Giuseppe Di Martino, Pamela Di Giovanni, Fabrizio Cedrone, Michela D’Addezio, Francesca Meo, Piera Scampoli, Ferdinando Romano, Tommaso Staniscia

**Affiliations:** 1Department of Medicine and Ageing Sciences, “G. d’Annunzio” University of Chieti-Pescara, 66100 Chieti, Italy; 2Unit of Hygiene, Epidemiology and Public Health, Local Health Authority of Pescara, 65100 Pescara, Italy; 3Department of Pharmacy, “G. d’Annunzio” University of Chieti-Pescara, 66100 Chieti, Italy; 4School of Hygiene and Preventive Medicine, “G. d’Annunzio” University of Chieti-Pescara, 66100 Chieti, Italy; 5Unit of Hygiene, Epidemiology and Public Health, Local Health Authority of Lanciano-Vasto-Chieti, 66100 Chieti, Italy; 6Department of Infectious Diseases and Public Health, “La Sapienza” University of Rome, 00185 Rome, Italy

**Keywords:** hip fracture, mortality, prediction score, HDR, Italy

## Abstract

*Background and Objectives*: The improved life expectancy was associated to the increased in the incidence of hip fractures among elderly people. Subjects suffering hip fractures frequently show concomitant conditions causing prolonged lengths of stay and higher in-hospital mortality. The knowledge of factors associated to in-hospital mortality or adverse events can help healthcare providers improve patients’ outcomes and management. The aim of this study was to develop a score to predict in-hospital mortality among hip fractured patients. *Materials and Methods*: Cases were selected from hospital admissions that occurred during the period 2006–2015 in Abruzzo region, Italy. The study population was split into two random samples in order to evaluate the accuracy of prediction models. A multivariate logistic regression was performed in order to identify factors associated to in-hospital mortality. All diagnoses significantly associated to in-hospital mortality were included in the final model. *Results*: The PRIMOF ranged between 0 and 27 and was divided into four risk categories to allow the score interpretation. An increase in odds ratio values with the increase in PRIMOF score was reported in both study groups. *Conclusions*: This study showed that a simple score based on the patient’ clinical comorbidities was able to stratify the risk of hip-fractured patients in terms of in-hospital mortality.

## 1. Introduction

The improved life expectancy was associated to the increase in the incidence of hip fractures among elderly people. Patients suffering hip fractures frequently show concomitant conditions causing prolonged lengths of stay and higher in-hospital mortality [1]. Hence, hip fractures in the geriatric population constitute a significant global public health issue [2,3]. The knowledge of factors associated to in-hospital mortality or adverse events can help healthcare providers improve patients’ outcomes and management [4]. In the scientific literature, various scoring tools have been developed to predict in-hospital mortality, though it is uncertain which of these is the most useful for patients with hip fractures. In particular, most of them—i.e., the Charlson Comorbidity Index [5] or POSSUM score [6]—have been adapted for surgical risk stratification, but none of them were specifically developed for hip-fractured patients. Some tools have been developed for hip-fractured patients [7], but they are frequently not easy to use. The ideal risk scoring should be simple, user-friendly, reproducible, and available to all patients, as reported by Jones et al. [8]. To this extent, the aim of this study was the development and validation of the “PRedict In-hospital MOrtality of hip Fractured” (PRIMOF) score, a predictive model derived from hospital discharge records. The secondary aim was to compare it with the Charlson Comorbidity Index (CCI), one of the most common tools used in risk stratification, in terms of discrimination.

## 2. Materials and Methods

### 2.1. Data Source

The study was conducted in Abruzzo, a Southern Italian region [9]. Patients were chosen from the hospital discharge record (HDR) referred to admissions that occurred during years 2006–2015. HDR included information about the demographic characteristics of each patient, a Diagnosis Related Group code (DRG) used to classify the hospitalization, and up to six possible diagnoses (one principal diagnosis and five possible comorbidities) and up to six possible procedures performed during the hospital stay, coded according to the International Classification of Disease, 9th Revision, Clinical Modification (ICD-9-CM).

### 2.2. Inclusion Criteria

All patients aged over 40 years admitted in the Abruzzo region for HF were included in the study. Only admissions reporting the codes from 820.0 to 820.9 (hip fracture) in the HDR as their diagnosis were included in the analysis. Additionally, the most frequent comorbidities reporting a prevalence of at least 1.5% and all diseases included in the CCI were collected and identified using the ICD-9-CM codes. All diagnoses were extracted and coded in accordance with the method proposed by Quan et al. [10].

### 2.3. Study Design

The study population was divided into two different random samples in order to evaluate the accuracy of predictions and to improve the reliability of all statistical models (Figure 1):−Training set, comprising about 50% of the subjects. All diagnoses were included in a logistic model to develop the score;−Validation set, comprising the second half of the sample. Here, the predictive properties of the score were validated.

### 2.4. Model Building and Statistical Analysis

Baseline information of included patients was reported as a frequency and percentage. Categorical variables were compared with a Chi-square test or Fisher’s exact test, where appropriate. A multivariate logistic regression model including all diagnoses was developed to identify diagnoses associated to in-hospital mortality. All diagnoses that resulted significant were included in the final model. A weight assigned to each variable was obtained from the regression coefficient value divided by 0.3; the value obtained was rounded to the nearest integer, as proposed by Gagne et al. [11]. The PRIMOF score was calculated from the overall sum of weights. Accurate predictions discriminate between those with and those without the study outcome. Discrimination power was evaluated by estimating the C index with the 95% confidence interval. Spearman’s correlation coefficient was estimated to evaluate the correlation between the CCI and PRIMOF score. Statistical significance was set to *p* < 0.05. Statistical analysis was performed by IBM SPSS Statistics v20.0 software (SPSS Inc. Chicago, IL, USA).

## 3. Results

During the study period, 23,075 hip-fractured patients were admitted to hospital in the Abruzzo region. Their median age (IQR) was 80.5 (69.8–89.8), and 16,749 were females (72.6%). During the hospitalization, 878 patients (3.8%) died. The training and validation samples included 11,477 and 11,598 subjects, respectively (Figure 1).

There were no significant differences between the two study groups in terms of baseline characteristics, as reported in Table 1.

### 3.1. Development of PRIMOF in the Training Sample

In the training set, a logistic model was developed to assess the weights of different diagnoses. All diagnoses not significantly associated with the study outcome were excluded from the final model. Table 2 reported all regression coefficient values calculated on in-hospital mortality and relatives assigned weight.

PRIMOF was obtained through the sum of the different diagnosis weights. The score ranged between 0 and 45. The highest score observed among enrolled patients in the training set was 27. Due to the small number of cases, subjects with a score higher than 10 were grouped and then considered in the upper risk class. After, 12 classes of PRIMOF score were identified. The score was divided into four groups to improve the score interpretation:−Class 1 from 0 to 3;−Class 2 from 4 to 6;−Class 3 from 7 to 9;−Class 4 higher than 9.

### 3.2. Validation Procedure and Comparison with the Charlson Comorbidity Index

A significant increase in odds ratio was reported with the increase in PRIMOF score in the validation group, closely emulating the results showed in the training set (Table 3).

The evaluation of accuracy was estimated via C-statistic (0.743; 95% CI 0.726–0.760; *p* < 0.001). The score reported a good calibration, with a non-significant Hosmer–Lemeshow test (Chi-squared 5.93; *p* = 0.313). PRIMOF and CCI were significantly correlated (Rho: 0.651; *p* < 0.001); however, compared to PRIMOF, CCI showed less accuracy in predicting in-hospital mortality in this sample. In particular, the C-index was 0.690 (95% CI 0.672–0.708; *p* < 0.001), as shown in Figure 2.

## 4. Discussion

### 4.1. Main Findings

This study demonstrates that a simple tool, based on clinical characteristics of patients, was able to stratify the risk of hip fractured population in terms of hospital mortality. In particular, the PRIMOF score performance was better than the CCI in predicting mortality among hip-fractured patients.

### 4.2. Comparison with Previous Studies

Many clinical risk scores were developed to predict in-hospital death of hip-fractured patients. There is a large variety of predictive tools used to identify patients at high risk through the mix of clinical and laboratory variables [12,13]. Frequently, obtaining all this information is unfeasible in many instances. Validated comorbidity indexes have also been used on admission data to predict the risk of mortality, but they were developed taking into account the general population and not hip-fractured patients. Particularly, many common scoring systems were adapted to hip fracture: ASA [14,15], CCI [16,17], and Nottingham Hip Fracture Score (NHFS) [7,17] are the most common. Additionally, the orthopedic version of POSSUM [6] was used for this aim, with Area Under the Receiver Operating Characteristic (AUROC) values ranging from 0.62 to 0.74 for mortality [12]. However, calibration showed poor results, with the observed and expected ratio ranging from 0.12 to 1.19 [12]. The NHFS and CCI use readily available pre-operative characteristics. They both have practical discriminant characteristics for mortality and were externally validated from their original cohort. The CCI was derived from well-defined variables. It is a moderately discriminant score for in-hospital morbidity and one-year mortality. However, calibration is not well reported, and this limits its capacity as audit tool. The NHFS is a hip-fracture-specific tool developed for early hospital discharge [18], 30-day mortality [7], and 1-year mortality [19]. Its discrimination ability is moderate and has good calibration. All the required items are routinely collected. Its main limitation is the use of the Abbreviated Mental Test Score (AMTS) as a cognitive impairment assessment score, which is not frequently used outside the UK [10]. In addition, NHFS did not predict in-hospital mortality.

### 4.3. Implications for Clinical Practice

The use of risk scores during the admission period has normally been accepted by physicians. It aims to improve clinical outcomes and service quality. The availability of a specific-disease tool able to catch the clinical complexity of admitted patients could help physicians in their daily activity. Moreover, the selection of patients at high risk of in-hospital mortality is useful for providers and healthcare services [20,21].

### 4.4. Strengths and Limitations

The main strength of the PRIMOF score is its reliance on a single data source, based on ICD-9-CM codes, widely used in many countries. In addition, it results in good accuracy and calibration.

The study has some limitations. Firstly, the identification of diagnosis is based on ICD-9-CM codes that do not consider the severity of reported diseases. Second, the use of HDR may be limited by the lack of certain types of information such as drug therapy. Finally, the true prevalence of some comorbidities was underestimated due to underreporting of them in the HDR.

## 5. Conclusions

This study showed that a simple score, based on the subject clinical history, can stratify the risk of in-hospital mortality among hip-fractured patients.

## Figures and Tables

**Figure 1 medicina-58-01082-f001:**
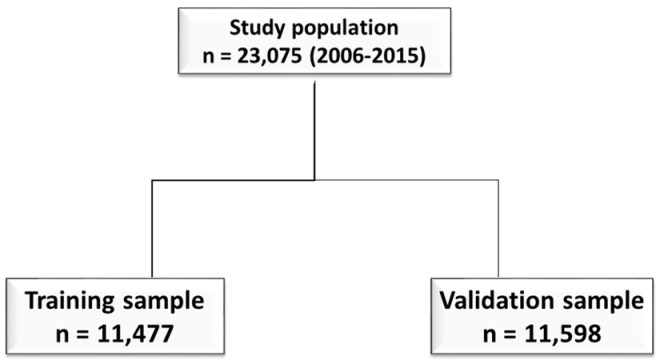
Flowchart of the study population.

**Figure 2 medicina-58-01082-f002:**
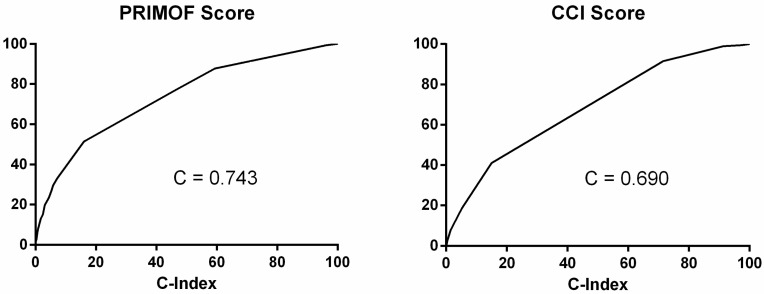
C-index of PRIMOF score and CCI score.

**Table 1 medicina-58-01082-t001:** Patient characteristics.

	Training Sample (n = 11,477)	Validation Sample (n = 11,598)	*p*-Value
Age			0.817
<65	905 (7.9)	935 (8.1)	
65–85	5822 (50.7)	5899 (50.9)	
>85	4745 (41.3)	4756 (41.0)	
Female gender	8320 (72.5)	8429 (72.7)	0.755
Italian	11,311 (98.6)	11,425 (98.5)	0.835
Public Hospital	10,441 (91.0)	10,545 (90.9)	0.785
Death	433 (3.8)	445 (3.8)	0.799
Cancer	110 (1.0)	105 (0.9)	0.674
Hematologic disease	2623 (22.9)	2646 (22.8)	0.942
Ischemic heart disease	101 (0.9)	106 (0.9)	0.785
Atrial fibrillation	277 (2.4)	252 (2.2)	0.222
Dementia	305 (2.7)	328 (2.8)	0.428
COPD	269 (2.3)	230 (2.0)	0.060
Heart failure	248 (2.2)	235 (2.0)	0.475
Mild Diabetes	750 (6.5)	732 (6.3)	0.489
Uncontrolled Diabetes	179 (1.6)	183 (1.6)	0.911
Peripheral vascular disease	24 (0.2)	29 (0.3)	0.516
Cerebrovascular disease	253 (2.2)	239 (2.1)	0.450
Rheumatologic disease	31 (0.3)	32 (0.3)	0.933
Ulcer	29 (0.3)	21 (0.2)	0.242
Slight hepatic disease	98 (0.9)	83 (0.7)	0.234
Severe hepatic disease	4 (0.1)	6 (0.1)	0.988
Plegia	63 (0.5)	52 (0.4)	0.278
Kidney disease	263 (2.3)	306 (2.6)	0.151
Metastasis	30 (0.3)	30 (0.3)	0.999
HIV/AIDS	1 (0.0)	3 (0.0)	0.989

**Table 2 medicina-58-01082-t002:** Assignment of weights in the development of PRIMOF score through multivariable logistic regression.

Diagnosis	Coefficient	OR (95% CI)	Weight
Age			
65–85	1.18	3.25 (1.58–6.68)	4
>85	2.09	8.06 (3.94–16.51)	7
Female gender	0.66	1.93 (1.57–2.39)	2
Cancer	1.25	3.51 (2.03–6.06)	4
Atrial fibrillation	0.52	1.68 (1.07–2.62)	2
COPD	0.58	1.79 (1.16–2.75)	6
Cerebrovascular disease	0.95	2.59 (1.67–4.01)	9
Ulcer	1.65	5.22 (1.74–15.71)	5
Kidney disease	0.95	2.58 (1.71–3.88)	3
Heart failure	1.97	7.18 (5.18–9.97)	7

**Table 3 medicina-58-01082-t003:** Odds ratios for in-hospital mortality according to PRIMOF score.

	Training Set		Validation Set	
Score	OR (95% CI)	*p*-Value	OR (95% CI)	*p*-Value
0–3	ref		ref	
4–6	3.35 (1.22–9.17)	0.018	2.61 (1.21–5.62)	0.014
7–9	9.98 (3.70–26.95)	<0.001	5.71 (2.68–12.19)	<0.001
>9	40.28 (14.85–109–24)	<0.001	23.99 (11.16–51.60)	<0.001

## Data Availability

Restrictions apply to the availability of these data. Data was obtained from Department of Health and Welfare of the Abruzzo region, Italy, and are available after reasonable request with the permission of the Department of Health and Welfare of the Abruzzo region.

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
