# Peer review of "Development and Validation of a New Tool in Predicting In-Hospital Mortality for Hip-Fractured Patients: The PRIMOF Score"

_medicina, 2022, doi:10.3390/medicina58081082_

Round 1
Reviewer 1 Report
Dear Authors:
I want to congratulate You on Your work.
It is an important point, no real clinical utility in my opinion (we have to treat these patients with a hip fracture, the only way to guarantee a positive results) but it is very interesting and useful in research and for epidemiology.
I do have some issues that I would like You to address. In particular, in my opinion, it should be better presented the score:
You wrote score from 0 to 27, but the sum of eights is 49: please explain.
You wrote that higher than 10 is upper risk class, but with the weights You presented a 85yo lady with minor comorbidity is at high risk: please explain.
How is "death" a baseline characteristic? (table 1)
Table 2 has a very short list of pathologies compared to Table 1: all the other pathologies were not important in stratifying the risk?
best regards,
Author Response
1) You wrote score from 0 to 27, but the sum of eights is 49: please explain.
Thank you for your observation. The sum of all heights is 45 (the age can be counted only once). Among enrolled patients, the highest score observed was 27. We corrected the sentence and clarified this point.
2) You wrote that higher than 10 is upper risk class, but with the weights You presented a 85yo lady with minor comorbidity is at high risk: please explain.
Thank you for your comment. The grouping procedure was based on the distribution of the score. Risk class were divided by quartile.
3) How is "death" a baseline characteristic? (table 1)
Following the reviewer’s suggestion, we have modified the title of Table 1.
Reviewer 2 Report
From my point of view, the inclusion criteria are not very well described. Primo-f score is not detailed. Correction of expression and spelling mistakes in English.
Author Response
- From my point of view, the inclusion criteria are not very well described.
Thank you for your observation. We improved the description of inclusion criteria and we added the section 2.2.
- Primo-f score is not detailed.
Thank you for your comment. The PRIMOF score was detailed in the section 2.4 – “Model building and statistical analysis”.
- Correction of expression and spelling mistakes in English.
Thank you for your comment. The English language was revised and all spelling mistakes were corrected.
This manuscript is a resubmission of an earlier submission. The following is a list of the peer review reports and author responses from that submission.